# Discordance, accuracy and reproducibility study of pathologists' diagnosis of melanoma and melanocytic tumors

Sarah Haggenmüller[1], Christoph Wies [1,2], Julia Abels[1], Jana T. Winterstein [1,2], Lukas Heinlein [1,2], Carina Nogueira Garcia[1], Jochen S. Utikal [3,4,5], Sebastian A. Wohlfeil [3,4,5], Friedegund Meier [6], Sarah Hobelsberger[6], Frank F. Gellrich [6], Mildred Sergon[7,8], Axel Hauschild[9], Lars E. French [10,11], Lucie Heinzerling[10,12], Justin G. Schlager[10], Kamran Ghoreschi [13], Max Schlaak [13], Franz J. Hilke[13], Gabriela Poch[13], Sören Korsing[13], Cosimo Sarfert[13], Carola Berking [12], Markus V. Heppt [12], Michael Erdmann [12], Sebastian Haferkamp[14], Konstantin Drexler[14], Dirk Schadendorf [15], Wiebke Sondermann [15], Matthias Goebeler [16], Bastian Schilling [17], Jakob Nikolas Kather [18], Stefan Fröhling [19], Mar Llamas-Velasco[20], Luis C. Requena[21], Gerardo Ferrara [22], Maite Fernandez-Figueras [23], Sylvie Fraitag[24], Cornelia S. L. Müller[25,26], Hans Starz[27], Heinz Kutzner[28], Raymond Barnhill[29], Richard Carr[30], Kenneth S. Resnik[31], Stephan Alexander Braun [32], Tim Holland-Letz[33] & Titus J. Brinker [1] ✉

Accurate melanoma diagnosis is crucial for patient outcomes and reliability of AI diagnostic tools. We assess interrater variability among eight expert pathologists reviewing histopathological images and clinical metadata of 792 melanoma-suspicious lesions prospectively collected at eight German hospitals. Moreover, we provide access to the largest panel-validated dataset featuring dermoscopic and histopathological images with metadata. Complete agreement is achieved in 53.5% of cases (424/792), and a majority vote (≥ five pathologists) in 90.9% (720/792). Considerable discordance is observed for non-invasive melanomas (complete agreement in only 10/73 cases). The expert panel disagrees with the local pathologists' and dermatologists' diagnoses in 14.9% and 33.5% of cases, respectively. This variability highlights the diagnostic challenges of early-stage melanomas and the need to reconsider how ground truth is established in routine care and AI research. Including at least two pathologists or virtual panels may contribute to more consistent diagnostic results.

Melanoma ranks as the primary cause of mortality among skin cancer patients worldwide[1]. In 2020, ~325,000 new cases of melanoma were reported globally, and 57,000 deaths were attributed to the disease[2]. Increased exposure to ultraviolet radiation, a well-known risk factor for the development of melanoma, largely drives this trend[3]. Consequently, melanoma incidences are projected to surge, with experts predicting 510,000 new cases and 96,000 deaths in 2040[2].

While certain melanoma cases exhibit aggressive behavior early on, the probability of metastasis increases significantly with Breslow thickness (BT)[4]. Detecting melanoma in its early stages thereby substantially increases the survival rates of affected patients. Therefore,

prompt and accurate identification of melanoma carries unprecedented importance at both the dermoscopic and (histo-)pathological levels.

Early diagnosis, however, remains challenging due to the frequent morphological overlap between melanoma and melanocytic nevi[5]. Despite technical advancements using dermoscopy, even experienced dermatologists rarely achieve sensitivity levels exceeding 80%[6]. In cases where clinical suspicion of malignancy cannot be ruled out by dermoscopic examination, a skin biopsy (excisional or incisional) is routinely performed. The histopathological examination of the biopsied lesion by a (dermato-)pathologist is considered the gold standard for diagnosing skin cancer. However, even this histopathological verification can yield inconclusive results, especially in borderline cases or thin melanomas (BT ≤ 1.0 mm[7]), with the latter constituting the majority of detections in skin cancer screenings[8,9].

Previous studies have revealed an interrater variability among (dermato-)pathologists of 17.2% to 26.0% in differentiating melanomas from nevi based on histopathological slides and associated clinical metadata[10–12]. These studies, however, often relied on single-center designs, utilized retrospectively collected data exclusively, relied on a limited number of cases, and/or did not consider non-invasive melanomas (NIMs) as a separate diagnostic category[10–16].

In response, we established an international expert pathologist panel to quantify the interrater reliability for skin lesions suspected of being melanoma, categorizing them as invasive melanomas (IMs), NIMs, nevi, and a grouped category for other diagnostic outcomes using a large prospectively and consecutively collected dataset from eight German university hospitals. In addition, we are providing access to this panel-validated dataset containing dermoscopic and histopathological whole-slide images, along with detailed lesion- and patient-specific metadata. By doing so, we aim to facilitate further (skin) cancer research and improve patient outcomes, while also advancing the reliability and development of artificial intelligence (AI)-driven diagnostic tools. These tools are increasingly demonstrating their effectiveness on both the dermoscopic (e.g., refs. 17,18) and (histo-)pathological level (e.g., refs. 19,20).

## Results

### Patient characteristics of the study sample and provided dataset

A total of 792 eligible slides from 736 patients were analyzed and are included in the provided dataset. Eight slides had to be excluded post-hoc due to inconsistencies in data collection or data quality. Patients' age at diagnosis ranged from 18 to 95 years, with a median patient age of 63 years. While all Fitzpatrick skin types were included in the prospective data collection, types II and III were the most prevalent, accounting for 57.5% and 28.5% of all patients, respectively. The most frequently affected body areas were the back (29.2% of all lesions) and sun-exposed regions such as lower extremities (19.1%), face/scalp/neck (16.5%), and upper extremities (12.9%). For a detailed breakdown of patient and lesion characteristics, please refer to Tables 1 and 2.

### Expert pathologist panel

**Panel characteristics.** Between December 2022 and July 2023, an expert team of eight board-certified (dermato-)pathologists (M.L.V., L.R., G.F., M.F.F., S.F., C.M., H.S., and H.K.) − equally composed of four females and four males (self-reported) − independently reviewed the WSIs of all 792 eligible lesions. The pathologists had a median clinical experience of 34 years, with individual experiences ranging from 13 to 40 years. In clinical reality, they reviewed a median of 1150 lesions per month, with individual monthly case volumes ranging from 400 to 7000.

**Agreement and arbitration procedures.** Complete agreement among all eight expert pathologists was reached in diagnosing 97 cases of IMs, 10 cases of NIMs, 238 cases of nevi, and 79 other diagnostic outcomes.

**Table 1 | Patient characteristics of the study sample and provided dataset**

|  | Overall (n = 792) | IM (n = 199) | NIM (n = 73) | Nevus (n = 373) | Others (n = 147) |
|---|---|---|---|---|---|
| **Age at diagnosis (years)** | | | | | |
| < 35 | 100 (12.6) | 4 (2.0) | 2 (2.7) | 89 (23.9) | 5 (3.4) |
| 35–54 | 193 (24.4) | 34 (17.1) | 6 (8.2) | 134 (35.9) | 19 (12.9) |
| 55–74 | 277 (35.0) | 82 (41.2) | 33 (45.2) | 106 (28.4) | 56 (38.1) |
| > 74 | 222 (28.0) | 79 (39.7) | 32 (43.8) | 44 (11.8) | 67 (45.6) |
| **Sex (physician-reported)** | | | | | |
| Male | 462 (58.3) | 122 (61.3) | 42 (57.5) | 197 (52.8) | 101 (68.7) |
| Female | 330 (41.7) | 77 (38.7) | 31 (42.5) | 176 (47.2) | 46 (31.3) |
| **Fitzpatrick skin type** | | | | | |
| I | 65 (8.2) | 29 (14.6) | 5 (6.8) | 23 (6.2) | 8 (5.4) |
| II | 453 (57.2) | 123 (61.8) | 52 (71.2) | 214 (57.4) | 64 (43.5) |
| III | 228 (28.8) | 38 (19.1) | 16 (21.9) | 115 (30.8) | 59 (40.1) |
| IV | 15 (1.9) | 2 (1.0) | 0 | 7 (1.9) | 6 (4.1) |
| VI | 1 (0.1) | 0 | 0 | 1 (0.3) | 0 |
| Unknown | 30 (3.8) | 7 (3.5) | 0 | 13 (3.5) | 10 (6.8) |
| **Personal history of IM/NIM** | | | | | |
| Yes | 157 (19.8) | 19 (9.5) | 16 (21.9) | 93 (24.9) | 29 (19.7) |
| No | 594 (75.0) | 172 (86.4) | 55 (75.3) | 262 (70.2) | 105 (71.4) |
| Unknown | 41 (5.2) | 8 (4.0) | 2 (2.7) | 18 (4.8) | 13 (8.8) |
| **Family history of IM/NIM** | | | | | |
| Yes | 63 (8.0) | 8 (4.0) | 2 (2.7) | 47 (12.6) | 6 (4.1) |
| No | 671 (84.7) | 181 (91.0) | 66 (90.4) | 303 (81.2) | 121 (82.3) |
| Unknown | 58 (7.3) | 10 (5.0) | 5 (6.8) | 23 (6.2) | 20 (13.6) |

Diagnostic categories are determined based on the corresponding expert panel-validated majority vote.

This full agreement accounted for 424 out of 792 cases, corresponding to 53.5% of all cases. Furthermore, a majority vote (i.e., five to seven pathologists agreeing) was achieved for 65 cases of IMs, 57 cases of NIMs, 117 cases of nevi, and 57 other diagnostic outcomes, leading to a majority agreement by the expert panel in 296 of 792, or 37.4%, of all cases. Consequently, in 720 of 792, or 90.9% of all cases a majority agreement or complete agreement was observable.

In 72 out of 792 cases or 9.1%, a majority vote was not reached because four or fewer experts agreed. The determination of the ground truth label was then assigned by the most experienced pathologist (HK; determined by both years of experience and monthly case volume). This decision served as the final arbitration. For a detailed breakdown of the rating frequencies per diagnostic category, please refer to Table 3.

**Quantification of interrater variability.** The diagnosis of IMs and NIMs represented as a percentage of the total 792 slides reviewed by each pathologist, varied from 18.8% to 28.9% for IMs and from 4.7% to 18.6% for NIMs. Notably, the most experienced reviewer, defined by both years of experience and monthly case volume, exhibited the highest IM ratio and lowest NIM ratio.

The overall chance-corrected interobserver agreement, measured by Fleiss' Kappa (κ), was 0.701 (95% CI: 0.693–0.710), indicating substantial agreement among the expert pathologists according to the Landis and Koch guidelines[21]. However, interobserver agreement varied across diagnostic categories. Substantial agreement was observed for other diagnostic outcomes (κ = 0.807, 95% CI: 0.794–0.820), nevi (κ = 0.742, 95% CI: 0.729–0.755), and IMs (κ = 0.725, 95% CI: 0.712–0.738). In contrast, the agreement for NIMs was only marginally

**Table 2 | Lesion characteristics of the study sample and provided dataset**

|  | Overall (n = 792) | IM (n = 199) | NIM (n = 73) | Nevus (n = 373) | Others (n = 147) |
|---|---|---|---|---|---|
| Lesion localization |  |  |  |  |  |
| Face/ scalp/neck | 131 (16.5) | 23 (11.6) | 35 (47.9) | 24 (6.4) | 49 (33.3) |
| Palms/soles | 19 (2.4) | 4 (2.0) | 1 (1.4) | 11 (2.9) | 3 (2.0) |
| Upper extremities | 102 (12.9) | 33 (16.6) | 16 (21.9) | 36 (9.7) | 17 (11.6) |
| Lower extremities | 151 (19.1) | 44 (22.1) | 5 (6.8) | 81 (21.7) | 21 (14.3) |
| Back | 231 (29.2) | 61 (30.7) | 12 (16.4) | 127 (34.0) | 31 (21.1) |
| Abdomen | 64 (8.1) | 14 (7.0) | 0 | 42 (11.3) | 8 (5.4) |
| Chest | 71 (9.0) | 17 (8.5) | 4 (5.5) | 38 (10.2) | 12 (8.2) |
| Buttock | 11 (1.4) | 2 (1.0) | 0 | 9 (2.4) | 0 |
| Genitalia | 11 (1.4) | 1 (0.5) | 0 | 5 (1.3) | 5 (3.4) |
| Unknown | 1 (0.1) | 0 | 0 | 0 | 1(0.1) |
| BT (in mm)[a] |  |  |  |  |  |
| Tis | 73 (9.2) | NA | 73 (100.0) | NA | NA |
| ≤1.0 (pT1) | 103 (13.0) | 103 (51.8) | NA | NA | NA |
| >1.0 to 2.0 (pT2) | 29 (3.7) | 29 (14.6) | NA | NA | NA |
| >2.0 to 4.0 (pT3) | 30 (3.8) | 30 (15.1) | NA | NA | NA |
| >4.0 (pT4) | 37 (4.7) | 37 (18.6) | NA | NA | NA |
| Not applicable (NA) | 520 (65.7) | NA | NA | 373 (100.0) | 147 (100.0) |

[a]BT categories are reported based on the corresponding local pathologist's assessment in routine care. In cases of diagnostic disagreement with the panel decision, an independent board-certified dermatologist (CNG), who was not part of the expert panel and has three years of experience in dermatohistopathology, re-evaluated all lesions suggested for reclassification from NIM/nevus to IM (n = 13).

Diagnostic categories are determined based on the corresponding expert panel-validated majority vote.

moderate (κ = 0.428, 95% CI: 0.415−0.441); significantly lower than for all other diagnostic categories according to the percentile method[22]. For a detailed breakdown of the κ values across subgroup analysis, please refer to Supplementary Table 1.

Subgroup analysis focusing on BT highlighted particular interrater variability in diagnosing pT1 melanomas. This was evidenced by the substantial number of lesions (33.0% of pT1 IMs vs. < 3.5% for pT2/pT3/pT4 IMs) for which no majority vote among the expert panel could be reached (see Table 4).

### Comparison with local pathologists

The established panel-validated ground truth concurred with the local pathologist in clinical care in 674 of 792 cases (85.1%). However, in 118 cases (14.9%), a more thorough review by the expert pathologist panel suggested that the initial diagnosis should be relabeled. IMs were most frequently reconsidered in absolute numbers (53 times), with 27 cases reclassified as NIMs, 15 as nevi, and 11 as other diagnostic outcomes. Nevi followed with 40 relabelings: eight as NIMs, seven as IMs, and 25 as other diagnostic outcomes. In addition, NIMs were re-evaluated 21 times, resulting in eight cases being reclassified as nevi, six as IMs, and seven as other diagnostic outcomes. Remarkably, in 11 cases, the initial pathological diagnosis by the local pathologists (including seven IMs, two NIMs, and two nevi) diverged from the panel-validated ground truth labeling, although the panel reached a complete agreement with all experts independently agreeing on the same diagnosis (indicating

only one IM, three NIMs, one nevus, and six other diagnostic outcomes).

Subgroup analysis at the clinic level revealed that University Hospital 8 exhibited a higher concordance rate with the expert pathologist panel compared to other clinics. In this hospital, ambiguous pigmented lesions are often examined preoperatively using confocal laser microscopy, with relevant findings being shared with the pathology department. Cases identified as dysplastic nevus, NIM, or IM undergo routine immunohistochemistry, typically using markers such as SOX10 and MelanA. Moreover, the diagnosis of melanoma strictly follows a mandatory four-eyes principle. For a detailed breakdown of concordance rates across all participating university hospitals, please refer to Table 5.

### Concordance with local dermatologist's clinical impression

In our study, the local pathologist disagreed with the local dermatologist's clinical impression in 221 out of 792 cases (27.9%). Within these, 134 of 221 cases (60.6%) involved a clinical suspicion of melanoma (IM or NIM) by the dermatologist, which the local pathologist later dismissed, highlighting the extent of avoidable biopsies. Conversely, in 42 of 221 cases (19.0%), dermatologists provided a clinical non-melanoma primary diagnosis (i.e., biopsy performed to rule out melanoma-suspicion as a differential diagnosis), while the local pathologist identified melanoma (IM or NIM). In addition, in twelve cases (5.4%), dermatologists suspected NIM, while the local pathologist diagnosed IM. Furthermore, in 33 cases, the diagnosis shifted between nevus and other diagnostic outcomes, or vice versa.

Similarly, the expert pathologist panel disagreed with the local dermatologist's clinical impression in 264 of 792 cases (33.3%). In 168 of these 264 cases (63.6%), the local dermatologists suspected melanoma (IM or NIM), which the expert panel ruled out. Conversely, in 35 of 264 cases (13.3%), dermatologists provided a clinical non-melanoma primary diagnosis (i.e., biopsy performed to rule out melanoma-suspicion as a differential diagnosis), while the expert panel confirmed melanoma (IM or NIM). In addition, in nine cases (3.4%), dermatologists indicated NIM, while the expert panel suggested IM. Furthermore, in 52 cases, the diagnosis shifted between nevus and other diagnostic outcomes, or vice versa.

Notably, subgroup analysis revealed that the higher the dermatologists' trust in their own diagnosis, the lower the discordance rates with both the local pathologist and the expert pathologist panel (see Table 6).

## Discussion

In this study, we quantified the interrater reliability among pathologists in diagnosing skin lesions suspected of melanoma and compared the panel decisions with the local pathologists' and dermatologists' labeling established in routine care. In addition, we provide access to the prospectively and consecutively collected, panel-validated dataset utilized in our study to facilitate further skin cancer research. Our findings demonstrate considerable interrater variability among expert pathologists in the multiclass classification of IMs, NIMs, nevi, and other diagnostic outcomes. Notably, discordance rates were particularly pronounced for pT1-melanomas, highlighting substantial variability in the diagnosis of early-stage melanomas and NIMs.

By leveraging a large, prospective, and consecutive data collection from multiple centers, we ensured a comprehensive representation of melanoma-suspicious cases typically encountered in routine clinical practice. Our expert pathologist panel demonstrated considerable variability in the histopathological diagnoses across diverse diagnostic categories, achieving complete agreement of the eight reviewing pathologists in only 424 out of 792 cases, with Fleiss' kappa (κ) ranging from 0.428 to 0.807. Our findings not only confirm but also extend previous reports of interrater discordance, which were largely based on single-center designs, retrospective data with limited sample

**Table 3 | Rating frequencies per diagnostic category**

| Highest rating frequency | Overall (%) | IM (%) | NIM (%) | Nevus (%) | Others (%) |
|---|---|---|---|---|---|
| < 5 (i.e., no majority vote) | 72 (9.1) | 37 (18.6) | 6 (8.2) | 18 (4.8) | 11 (7.5) |
| 5 | 81 (10.2) | 14 (7.0) | 28 (38.4) | 26 (7.0) | 13 (8.8) |
| 6 | 73 (9.2) | 12 (6.0) | 15 (20.5) | 36 (9.7) | 10 (6.8) |
| 7 | 142 (17.9) | 39 (19.6) | 14 (19.2) | 55 (14.7) | 34 (23.1) |
| 8 | 424 (53.5) | 97 (48.7) | 10 (13.7) | 238 (63.8) | 79 (53.7) |
| Overall | 792 (100.0) | 199 (100.0) | 73 (100.0) | 373 (100.0) | 147 (100.0) |

For cases without a majority vote (less than 5 out of 8 agreeing experts), the tie-break decision based on the most experienced reviewer is reported.

**Table 4 | Rating frequencies for panel-assured IMs per BT category**

| Highest rating frequency | Overall number of IMs (%) | ≤1.0 (pT1)[a] | > 1.0–2.0 (pT2)[a] | > 2.0–4.0 (pT3)[a] | > 4.0 (pT4)[a] |
|---|---|---|---|---|---|
| < 5 (i.e., no majority vote) | 37 (18.6) | 34 (33.0) | 1 (3.4) | 1 (3.3) | 1 (2.7) |
| 5 | 14 (7.0) | 12 (11.7) | 1 (3.4) | 0 | 1 (2.7) |
| 6 | 12 (6.0) | 10 (9.7) | 1 (3.4) | 1 (3.3) | 0 |
| 7 | 39 (19.6) | 24 (23.3) | 4 (13.8) | 6 (20.0) | 5 (13.5) |
| 8 | 97 (48.7) | 23 (22.3) | 22 (75.9) | 22 (73.3) | 30 (81.1) |
| Overall | 199 (100.0) | 103 (100.0) | 29 (100.0) | 30 (100.0) | 37 (100.0) |

[a]BT categories are filtered based on the corresponding local pathologist's assessment in routine care. In cases of diagnostic disagreement with the panel decision, an independent board-certified dermatologist (CNG), who was not part of the expert panel and has three years of experience in dermatohistopathology, re-evaluated all lesions suggested for reclassification from NIM/nevus to IM (n = 13).

**Table 5 | Subgroup analysis for comparison with the local pathologists at clinic level**

| | Disagreement with expert panel | Agreement with expert panel | Overall number of cases |
|---|---|---|---|
| Hospital 1 | 14 (17.5) | 66 (82.5) | 80 (100.0) |
| Hospital 2 | 33 (19.3) | 138 (80.7) | 171 (100.0) |
| Hospital 3 | – | – | 21 (100.0) |
| Hospital 4 | – | – | 21 (100.0) |
| Hospital 5 | 27 (18.1) | 122 (81.9) | 149 (100.0) |
| Hospital 6 | 9 (19.1) | 38 (80.9) | 47 (100.0) |
| Hospital 7 | 20 (18.3) | 89 (81.7) | 109 (100.0) |
| Hospital 8 | 11 (5.7) | 183 (94.3) | 194 (100.0) |
| **All hospitals** | **118 (14.9)** | **674 (85.1)** | **792 (100.0)** |

Participating university hospitals with fewer than 25 samples are excluded due to limited representativeness.

sizes, and focused on invasive melanomas[10–16], reporting interrater variability of 17.2% to 26.0% in differentiating melanomas from nevi based on histopathological slides and associated clinical metadata[10–12]. Notably, in our study, discordance was particularly pronounced in the diagnosis of NIMs ($\kappa$ = 0.428), with full agreement reached in only 10 out of 73 cases. This highlights the significant diagnostic challenges posed by premalignant stages, such as NIMs, even among experienced international experts.

Furthermore, our findings reveal that the dermatological primary diagnosis conflicted with the local pathologists' labeling and panel decisions in about 30% of all cases. This discrepancy resulted in substantial dermatological overdiagnosis (i.e., avoidable biopsies), affecting 60–65% of these cases. However, in dermatological practice, overdiagnosis is often influenced by additional factors, including patient concerns, which play a considerable role in clinical decision-making and should be considered when interpreting rates of overdiagnosis. Moreover, our comparison between the local pathologists' labeling and the panel decisions indicates that also at the pathological level diagnostic uncertainty may result in over- and misdiagnosis (27 and 31 cases of IMs/NIMs suggested for reclassification by the expert panel, respectively) as well as underdiagnosis (23 cases of NIMs/ nevi recommended for relabeling). Remarkably, in 11 cases (including nine initial IM/NIM cases), the local pathologists diverged from the panel-validated ground truth labeling, although the expert panel achieved a complete agreement (i.e., all eight reviewers agreeing independently). The outlined potential over- and misdiagnosis not only places considerable physical and psychological burdens on the patients themselves[23] but also affects their families and partners[24], while concurrently generating avoidable healthcare costs. At the same time, underdiagnosis remains a major concern, with numerous patients potentially not receiving timely and appropriate treatment for melanoma. To mitigate these severe consequences, it would be advisable that at least two pathologists independently assess each case, with the provision to request additional staining or consult a third party, such as an independent expert review board (e.g., ref. 25,), if discrepancies arise. However, in clinical practice, capacity constraints and a shortage of (dermato-)histopathologists may render these recommendations unfeasible.

In these situations, AI-based diagnostic tools have emerged as promising support systems[19,20,26], as evidenced in a large prospective breast cancer study[27]. Here, AI-supported screening matched cancer detection rates of the standard double readings performed by radiologists, but with a substantially reduced screen-reading workload, thereby highlighting the potential of AI to replace radiologists in the role of double-checking diagnostic images[27]. However, the effectiveness of such AI systems hinges on the quality of the underlying training data used to develop these models. If neural networks are trained with inaccurately labeled images during supervised learning, these errors are assimilated into the model-building process[28,29]. This can lead to diminished classification performance[29] and potentially jeopardize patient safety, highlighting the urgent need for datasets with highly reliable ground truth labels.

Consequently, accurately diagnosing melanoma-suspicious skin lesions is critical for multiple reasons. It not only mitigates the prevalent issue of over- and underdiagnosis — particularly in early-stage melanomas which constitute the majority of detections in skin cancer screenings[8,9] — but also ensures consistent diagnostic quality and reduces the risk of incorporating misdiagnoses in the development

**Table 6 | Subgroup analysis on dermatologists' diagnostic confidence**

| Diagnostic confidence | Disagreement with local pathologist | Agreement with local pathologist | Disagreement with expert panel | Agreement with expert panel |
|---|---|---|---|---|
| 1 | 5 (35.7%) | 9 (64.3%) | 10 (71.4%) | 4 (28.6%) |
| 2 | 31 (52.5%) | 28 (47.5%) | 31 (52.5%) | 28 (47.5%) |
| 3 | 117 (35.8%) | 210 (64.2%) | 135 (41.3%) | 192 (58.7%) |
| 4 | 52 (20.6%) | 201 (79.4%) | 63 (24.9%) | 190 (75.1%) |
| 5 | 15 (11.03%) | 121 (89.0%) | 25 (18.4%) | 111 (81.6%) |
| Overall[a] | 220 (27.9%) | 569 (72.1%) | 264 (33.5%) | 525 (66.5%) |

[a]The diagnostic uncertainty for three out of 792 cases was labeled as unknown and therefore excluded for the comparison.
1 indicates low diagnostic confidence, whereas 5 indicates high diagnostic confidence.

process of innovative diagnostic tools. This emphasizes the urgent need to rethink how ground truth is established in both routine patient care and future AI research. Routinely involving at least two pathologists or using virtual panels in the diagnostic process of ambiguous cases may lead to more consistent diagnostic results (see concordance rate of hospital 8) − particularly for diagnosing NIMS and thin melanomas. In addition, employing molecular analyses and expanding the use of immunohistochemical staining (as done at hospital 8) could further enhance diagnostic consistency and reliability.

Although our study design enabled pathologists to review the lesions alongside patient- and lesion-specific metadata − thus creating a setting closely mimicking clinical practice[30,31] − the expert pathologist panel lacked access to immunohistochemistry staining or additional preoperative information, such as clinical images of the lesions[32,33] or findings from confocal microscopy, which could have further enhanced diagnostic accuracy. Consequently, while the labels were established by an independent expert pathologist panel − surpassing the conventional standard of care that typically involves histopathological verification by a single pathologist − interpreting the interrater agreement requires caution. In addition, we acknowledge that variations in how clinical information was accessed may have contributed to diagnostic differences, and there might be considerable variations in dermatopathology practices across geographic regions and between academic institutions and private practice. These factors should be considered when interpreting the interrater agreement and the broader applicability of our findings.

Considerable interrater variability among expert pathologists highlights the diagnostic challenges of thin and non-invasive (in situ) melanomas, frequently encountered in early detection settings such as skin cancer screenings. This discordance may lead to numerous healthy individuals being misdiagnosed and unnecessarily treated for melanoma. Conversely, a substantial number of patients may not receive appropriate treatment due to over-, mis- or underdiagnosis, potentially resulting in worse patient outcomes. Routinely involving virtual panels in the diagnostic process may contribute to more consistent diagnostic results, thereby supporting improved patient care.

## Methods
### Ethics statement and reporting standards
Ethics approval was obtained from the ethics committee at the Technical University of Dresden (BO-EK-53012021), the Friedrich-Alexander University Erlangen-Nuremberg (69_21 Bc), the LMU Munich (21-0182), the University of Regensburg (20-2190-103), the Julius-Maximilians University Wuerzburg (293/20_z) and from the University Hospitals Mannheim (2020-656 N) and Essen (20-9784-BO). Patients provided informed written consent. The study was performed in accordance with the Declaration of Helsinki and followed the Standards for Reporting of Diagnostic Accuracy (STARD 2015) guidelines[34].

### Patient cohorts, image acquisition, and slide acquisition
Study participants were prospectively enrolled in the Skin Classification Project (SCP2) across eight university hospitals in Germany (Berlin, Dresden, Erlangen, Essen, Mannheim, Munich, Regensburg, Wuerzburg) from April 2021 to February 2023. The inclusion criteria required patients to be at least 18 years old and present with pigmented melanoma-suspicious skin lesions, either as a clinical primary diagnosis or as a secondary/differential diagnosis with the intent of ruling out melanoma. The exclusion criteria prohibited the enrollment of pre-biopsied lesions or lesions located under the finger- or toenails. In addition, patients with person-identifying features (e.g., tattoos) in the immediate vicinity of the lesions were excluded due to data privacy concerns.

Data on skin lesions clinically suspicious of melanoma, which were subsequently excised, were gathered prospectively and consecutively as part of routine clinical care. This comprehensive data collection included six dermoscopic images for each lesion, captured with variations in the orientation, position, and operational mode of the dermoscope (including both polarized and non-polarized settings). To minimize the impact of confounding variables, known artifacts, such as skin markings, were deliberately avoided during image acquisition. All images were taken using one out of four distinct hardware configurations that were consistently employed across the participating hospitals (HEINE Delta30 dermoscope with Apple iPhone 7, HEINE DeltaOne with Apple iPhoneSE, HEINE DeltaOne with Apple iPhone 8, and HEINE IC1 with Apple iPhone 7). In addition to the images, the dermatologist's primary diagnosis and diagnostic confidence on a scale from 1 (low) to 5 (high) was recorded. Moreover, detailed patient-specific information, such as age at diagnosis, sex (physician-reported), Fitzpatrick skin type, and personal and family history of melanoma, as well as lesion-specific data, including pigmentation, lesion localization, and diameter, were collected.

The diagnostic labels were subsequently verified through histopathological examination by at least one (dermato-)pathologist (referred to as a local pathologist) at the corresponding hospital, as part of routine clinical practice. In cases involving multiple tumors (collision cases), the label of the largest tumor region was assigned to clarify the diagnostic focus. In addition, a comprehensive set of histopathological parameters was recorded for each diagnostic category, including lesion subtypes. For melanoma, the AJCC stage, Clark level, BT, mitotic rate, ulceration, regression, and sentinel lymph node status were documented. For nevi, parameters such as the distribution of nevus cells and the presence of atypia were recorded.

Following the routine excision of the enrolled melanoma-suspicious lesions, the corresponding hematoxylin-eosin-stained (H&E) reference slides used for pathological diagnosis in clinical practice were centrally digitized at the German Cancer Research Center (DKFZ). A Leica Aperio AT2 DX slide scanner (Leica Biosystems) was utilized to digitize these slides at 40x magnification with a resolution of 0.25 µm/px.

## Expert-panel validated labeling

In addition, 18 international expert (dermato-)pathologists, unaffiliated with the participating university hospitals, were invited to participate in the study. Of these, 12 began the evaluation process and eight completed assessments of all 800 included lesions. The digitized H&E reference slides of all prospectively and consecutively collected lesions were provided in two batches using CytoBrowser, an online platform customized for our study. The digitized whole slide images (WSIs) were displayed in a random order. In addition, patient-specific (year of birth, age at diagnosis, sex, skin type according to Fitzpatrick, personal and family history of melanoma) and lesion-specific metadata (lesion localization) were displayed (see Supplementary Fig. 1).

Participating pathologists were assigned to classify each reviewed lesion into one of the respective diagnostic categories: IM, NIM (defined as melanoma in situ and lentigo maligna; Clark level I), nevus, or other diagnostic outcome. For further details, please refer to Supplementary Table 3. In addition, the pathologists had the opportunity to flag any issues with image/slide quality and to provide comments in a free-text format. The web-based environment supported functionalities such as zooming and rotating the WSIs for detailed diagnostic examination.

## Statistics

Fleiss' Kappa ($\kappa$) was employed to statistically quantify the level of agreement among the raters within the expert pathologist panel, whilst simultaneously accounting for any agreement that might occur by chance. $\kappa$ ranges from $-1$ to 1, where 1 represents perfect agreement, 0 denotes agreement expected by chance, and $-1$ indicates perfect disagreement. Higher values suggest greater agreement among raters, whereas negative values indicate systematic disagreement. The interpretation of $\kappa$ followed the Landis and Koch guidelines[21]. Confidence intervals (CI) were calculated using the bootstrap method. A significance level of alpha = 0.05 was set for the calculations of the CIs. Thus, we calculated 95% CIs and investigated their overlap to assess significant differences in $\kappa$ according to the percentile method[22]. All statistical analyses were performed using SPSS version 29.0.0.0 (IBM Corporation) and R version 4.1.2. The CIs assume independence of observations, which is slightly violated due to 54 patients contributing with two lesions to the dataset. To validate the credibility of our CIs despite this violation, we calculated $\kappa$ for the greatest independent dataset, ending up in a drop of only 0.011/0.001, indicating that the violation is negligible (see Supplementary Table 2).

## Reporting summary

Further information on research design is available in the Nature Portfolio Reporting Summary linked to this article.

# Data availability

External research projects may request access to the prospectively and consecutively collected, panel-validated dataset utilized in our study, specifically for the purpose of advancing skin (cancer) research. Access is granted following an application and approval process managed by the SCP Data Protection Committee, which evaluates requests based on criteria such as alignment with patient consent, a valid ethics vote, and other relevant requirements (i.e., non-commercial (skin) cancer research). All remaining data is available in the article, supplementary and source data files. Commercial use of the dataset is strictly prohibited.

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

## Acknowledgements

This study was funded by the Federal Ministry of Health, Berlin, Germany (grant: Skin Classification Project 2) and the Ministry of Health, Social Affairs and Integration Baden-Württemberg, Stuttgart, Germany (grant: sKIn). Grant holder in both cases: Titus J. Brinker, German Cancer Research Center, Heidelberg, Germany. The funders had no role in the design and conduct of the study; collection, management, analysis, and interpretation of the data; preparation, review, or approval of the manuscript; and decision to submit the manuscript for publication.

## Author contributions

Sarah Haggenmüller: conceptualization, data curation, formal analysis, investigation, methodology, project administration, software, visualization, writing – original draft. Christoph Wies: formal analysis, investigation, methodology, software, writing – original draft, writing – review & editing. Julia Abels: validation, visualization, writing – review & editing. Jana T. Winterstein: validation, visualization, writing – review & editing. Lukas Heinlein: validation, visualization, writing – review & editing. Carina Nogueira Garcia: validation, visualization, writing – review & editing. Jochen S. Utikal: resources, visualization, writing – review & editing. Sebastian A. Wohlfeil: resources, visualization, writing – review & editing. Friedegund Meier: resources, visualization, writing – review & editing. Sarah Hobelsberger: resources, visualization, writing – review & editing. Frank F. Gellrich: resources, visualization, writing – review & editing. Mildred Sergon: resources, visualization, writing – review & editing. Axel Hauschild: validation, visualization, writing – review & editing. Lars E. French: resources, visualization, writing – review & editing. Lucie Heinzerling: resources, visualization, writing – review & editing. Justin G. Schlager: resources, visualization, writing – review & editing. Kamran Ghoreschi: resources, visualization, writing – review & editing. Max Schlaak: resources, visualization, writing – review & editing. Franz J. Hilke: resources, visualization, writing – review & editing. Gabriela Poch: resources, visualization, writing – review & editing. Sören Korsing: resources, visualization, writing – review & editing. Cosimo Sarfert: resources, visualization, writing – review & editing. Carola Berking: resources, visualization, writing – review & editing. Markus V. Heppt: resources, visualization, writing – review & editing. Michael Erdmann: resources, visualization, writing – review & editing. Sebastian Haferkamp: resources, visualization, writing – review & editing. Konstantin Drexler: resources, visualization, writing – review & editing. Dirk Schadendorf: resources, visualization, writing – review & editing. Wiebke Sondermann: resources, visualization, writing – review & editing. Matthias Goebeler: resources, visualization, writing – review & editing. Bastian Schilling: resources, visualization, writing – review & editing. Jakob Nikolas Kather: resources, visualization, writing – review & editing. Stefan Fröhling: resources, visualization, writing – review & editing. Mar Llamas-Velasco: resources, visualization, writing – review & editing. Luis C. Requena: resources, visualization, writing – review & editing. Gerardo Ferrara: resources, visualization, writing – review & editing. Maite Fernandez-Figueras: resources, visualization, writing – review & editing. Sylvie Fraitag: resources, visualization, writing – review & editing. Cornelia S. L. Müller: resources, visualization, writing – review & editing. Hans Starz: resources, visualization, writing – review & editing. Heinz Kutzner: resources, visualization, writing – review & editing. Raymond Barnhill: resources, visualization, writing – review & editing. Richard Carr: resources, visualization, writing – review & editing. Kenneth S. Resnik: resources, visualization, writing – review & editing. Stephan Alexander Braun: resources, visualization, writing – review & editing. Tim Holland-Letz: conceptualization, methodology, validation, writing – review & editing. Titus J. Brinker: funding acquisition, project administration, conceptualization, supervision, validation, writing – review & editing.

## Funding

## Competing interests

Prof Utikal reported personal fees from Amgen, Bristol Myers Squibb, GSK, Immunocore, LEO Pharma, Merck Sharp & Dohme, Novartis, Pierre Fabre, Rheacell, Roche, and Sanofi outside the submitted work. Dr Wohlfeil received honoraria from Bristol Myers Squibb, Novartis, and Sun Pharma outside the submitted work. Prof Meier reported grants from Novartis and Roche and others (travel support or/and speaker's fees or/and advisor's honoraria) from BMS, MSD, and Pierre Fabre outside the submitted work. Dr Hobelsberger reported clinical trial support and advisor's honoraria from Almirall, speaker's honoraria from Almirall, UCB, and AbbVie, and travel support from UCB, Janssen Cilag, Almirall, Novartis, Lilly, LEO Pharma, and AbbVie outside the submitted work. Prof Heinzerling reported other (clinical studies) from BMS, MSD, Pierre Fabre, Replimune, and Sanofi; personal fees from Biomedx, BMS, MSD, Sun, Pierre Fabre, Novartis, and Sanofi; and grants from Therakos outside the submitted work. Prof Haferkamp reported speaker's fees and advisor's honoraria from BMS, MSD, Novartis, and Pierre Fabre outside the submitted work. Dr Schlaak reported personal fees from BMS, Novartis, Immunocore, Sun Pharma, MSD, Recordati, and Sanofi Aventis outside the submitted work. Prof Berking reported personal fees from BMS, MSD, InflaRx, Novartis, Sanofi, LEO Pharma, Almirall Hermal, Pierre Fabre, Immunocore, and Delcath outside the submitted work. Dr Erdmann received travel support and speaker's honoraria from Immunocore, Novartis, Pierre Fabre, and Sanofi outside the submitted work. Dr Sondermann reported grants from Almirall, Novartis, and Medi GmbH; and personal fees from AbbVie, Almirall, BMS, Boehringer Ingelheim, Celgene, Janssen, LEO Pharma, Lilly, Novartis, Pfizer, Sanofi Genzyme, and UCB outside the submitted work. Prof Goebeler reported grants for clinical studies from Argenx, Novartis, Janssen, and Galderma; personal fees from Almirall (consulting), Janssen (advisory board, speaker), GSK (advisory board, speaker), and Lilly

(speaker) outside the submitted work. Dr Llamas-Velasco reported fees as an advisory board member, consultant, research support, participation in clinical trials, and honorary for speaking with the following pharmaceutical companies: Abbvie, Almirall, Amgen, Boehringer, BMS, Celgene, Janssen, Leo, Lilly, Kyowa kirin, Novartis and UCB, outside the submitted work. Dr. Kather declares consulting services for Owkin, France; DoMore Diagnostics, Norway; Panakeia, UK; AstraZeneca, UK; Scailyte, Switzerland; Mindpeak, Germany; and MultiplexDx, Slovakia. Furthermore, he holds shares in StratifAI GmbH, Germany, has received a research grant from GSK, and has received honoraria from AstraZeneca, Bayer, Eisai, Janssen, MSD, BMS, Roche, Pfizer, and Fresenius. Dr Gabriela Poch received travel support and speaker's honoraria from MSD, BMS, Novartis, Sun Pharma, and Amgen outside the submitted work. Dr Brinker reported being the owner of Smart Health Heidelberg GmbH outside the submitted work. The remaining authors declare no competing interests.

## Additional information

[1]Digital Biomarkers for Oncology Group, German Cancer Research Center (DKFZ), Heidelberg, Germany. [2]Medical Faculty, University Heidelberg, Heidelberg, Germany. [3]Department of Dermatology, Venereology and Allergology, University Medical Center Mannheim, Ruprecht-Karl University of Heidelberg, Mannheim, Germany. [4]Skin Cancer Unit, German Cancer Research Center (DKFZ), Heidelberg, Germany. [5]DKFZ Hector Cancer Institute at the University Medical Center Mannheim, Mannheim, Germany. [6]Skin Cancer Center at the University Cancer Center and National Center for Tumor Diseases Dresden, Department of Dermatology, Faculty of Medicine and University Hospital Carl Gustav Carus, Technische Universität Dresden, Dresden, Germany. [7]Institute of Pathology, Faculty of Medicine and University Hospital Carl Gustav Carus, Technische Universität Dresden, Dresden, Germany. [8]Skin Cancer Center at the National Center for Tumor Diseases (NCT/UCC, Faculty of Medicine and University Hospital Carl Gustav Carus, Technische Universität Dresden, Dresden, Germany. [9]Department of Dermatology, University Hospital (UKSH), Kiel, Germany. [10]Department of Dermatology and Allergy, University Hospital, LMU Munich, Munich, Germany. [11]Dr. Phillip Frost Department of Dermatology and Cutaneous Surgery, University of Miami, Miller School of Medicine, Miami, FL, USA. [12]Department of Dermatology, University Hospital Erlangen, Comprehensive Cancer Center Erlangen – European Metropolitan Region Nürnberg, CCC Alliance WERA, Erlangen, Germany. [13]Department of Dermatology, Venereology and Allergology, Charité – Universitätsmedizin Berlin, Corporate member of Freie Universität Berlin and Humboldt-Universität zu Berlin, Berlin, Germany. [14]Department of Dermatology, University Hospital Regensburg, Regensburg, Germany. [15]Department of Dermatology, Venereology and Allergology, University Hospital Essen, Essen, Germany. [16]Department of Dermatology, Venereology and Allergology, University Hospital Würzburg and National Center for Tumor Diseases (NCT) WERA Würzburg, Würzburg, Germany. [17]Goethe University Frankfurt, University Hospital, Department of Dermatology, Frankfurt, Germany. [18]Else Kroener Fresenius Center for Digital Health, Technical University Dresden, Dresden, Germany. [19]Department of Translational Medical Oncology, National Center for Tumor Diseases (NCT) Heidelberg and German Cancer Research Center (DKFZ), Heidelberg, Germany. [20]Department of Dermatology, University Hospital La Princesa, Madrid, Spain. [21]Dermatology Department, Fundación Jiménez Díaz, Autonomous University of Madrid, Madrid, Spain; Anatomic Pathology Service, Fundación Jiménez Díaz, Madrid, Spain. [22]Anatomic Pathology and Cytopathology Unit, Istituto Nazionale Tumori I.R.C.C.S. Fondazione 'G. Pascale', Naples, Italy. [23]University General Hospital of Catalonia, Grupo Quironsalud, International University of Catalonia, Sant Cugat del Vallés, Barcelona, Spain. [24]Pathology department, Necker-Enfants Malades Hospital, Université Paris-Cité, Assistance Publique des Hopitaux de Paris, Paris, France. [25]Center for Histology, Cytology and Molecular Diagnostics, Trier, Germany. [26]Saarland University, Homburg/Saar, Germany. [27]Dermpath München, Munich, Germany. [28]Dermatopathology Friedrichshafen, Friedrichshafen, Germany. [29]Departments of Pathology and Translational Research, Institut Curie, Paris, France. [30]Department of Pathology, Warwick Hospital, Warwick, UK. [31]Institute for Dermatopathology, Newtown Square, PA, USA. [32]Department of Dermatology, University of Münster, Münster, Germany; Department of Dermatology, Medical Faculty, Heinrich-Heine-University, Düsseldorf, Germany. [33]Department of Biostatistics, German Cancer Research Center (DKFZ), Heidelberg, Germany. ✉e-mail: titus.brinker@dkfz.de

