## [Transparent Peer Review file · Nature Communications]

Discordance, Accuracy and Reproducibility Study of Pathologists' Diagnosis of Melanoma and Melanocytic Tumors

Corresponding Author: Dr Titus Brinker

Version 0:

Reviewer comments:

Reviewer #1

(Remarks to the Author)

Congratulations and thank you to the authors and team for this important work that addresses a foundational issue at the heart of AI capabilities as applied to pigmented lesion (an in fact more broadly to skin lesion) classification.

The team here expands on the broadly-recognized concept pertaining to the question of the quality of the gold standard of histopathologic diagnosis of melanocytic lesions. It has been well-demonstrated that there is heterogeneity in both inter and intra-reader histopathologic diagnosis, that raises questions about the implications for work in the development of AI tools and capabilities that rely on this gold standard labeling. Further, this topic and issue further plays into current controversies about diagnostic shift and the role of histopathologic diagnosis may play in the rising incidence of melanoma and melanoma-in-situ diagnosis in the absence of rising mortality (i.e. the question of an epidemic driven by over diagnosis/over-surveillance)?

This study further adds to this body of literature with important additions given the robust study design that looks across multiple centers, separates out invasive from in-situ melanoma, and utilized a prospective design, in addition to the valuable sharing of a robust prospectively-collected multi-modal dataset consisting of deeply clinically annotated dermoscopic and histopathologic images.

In terms of study design, how does the review of WSI, while increasingly prevalent, differ from usual practice of the local dermatopathologist who may have access to the entire block and additional slides and special stains? In these cases, were only H&E sections reviewed, or was there use of special stains (PRAME, etc?).

Were the panelists restricted to pre-specified categories of histopathologic diagnosis? What categories are captured in the "other" vs. nevus category? In other words, how were atypical spitz, MELTUMP, mild/mod/severely atypical nevi, etc. categorized--all together in "other"?

Very interesting insight that institution 8 has a number of structural factors pertaining to their dermatopathology practice that looks to promote diagnoses more concordant to expert consensus, specifically use of non-invasive imaging capabilities wielded in clinical setting, use of immunohistochemistry, and consensus conference/4 eyes principle. Would be interested in which if these approaches is used at the other institutions (non-invasive diagnostics clinically, special staining, and consensus conference or approach for challenging lesions or IMs).

What was the nature of the comparison of pathologist with dermatologist? Top 1? Top 3? Free text or choosing from pre-specified list? How might a situation, for instance, of a clinician who was moderately concerned about MIS but top 1 diagnosis was moderately or severely atypical nevus be treated? Top 1 being atypical nevus but low confidence because Top 2 would include MIS? Wondering whether the approach of concern for melanoma (very high, high, moderate, low, very low) might be a more appropriate approach.

It looks like a standardized constructed was created to capture clinical information; was this same information provided in the same way to local pathologist, or was this just for the expert group? Wondering whether these difference may have confounded the differences between local and expert pathologic diagnoses. Appreciate the authors acknowledgment of these differences and variations in the manuscript section on limitations. I believe that there are also significant variations in

dermpath practices geographically and between academic institutions and private practice, perhaps noting these additional limitations of the broader applicability of these findings might be appropriate.

The framing of "Comparison with Local Dermatologist" is not an apples to apples (clinical diagnosis with path diagnosis) and may be better converted as the concordance between clinician impression and histopathologic diagnosis. As we know, the diagnosis is one of clinical-pathologic correlation where the two are complementary approaches. I would expect and how that there would be considerable difference in a top 1 clinical dx and pathologic diagnosis where the certainty of the diagnosis moves with concordance with pathologic diagnosis. What I think would be more interesting is level of suspicion or certainty of diagnosis of melanoma/MIS.

Page 17 last paragraph speaks to the discordance between local read and expert read with the majority of instances representing an "overcall" from local pathologist. Perhaps would soften the language re: "substantial overdiagnosis" and perhaps just have the proportion of over/under call stand alone and the line about "suggestion that approximately 10% of all melanoma cases might be pathologically overdiagnosed". I would also argue that deserving similar highlighting is the proportion of cases that may be pathologically under diagnosed though less frequent or common.

I am curious as to whether the authors may perform an analysis that might support their recommendation of 2 pathologist independently assess a case (i.e. what would the data have looked like randomly choosing a pair of the expert reviewer panel vs. the full 8)? How does the practice of local consensus conference or 4 eye approach potentially compare or address this in the setting of resource constraint?

Appreciate the paradox called out in the discussion section where we have a real-world clinical opportunity of generating more consistent and accurate pathologic diagnoses and AI can potentially serve as a technological layer to help, but that is predicated on more robust ground-truth labeling. I wonder whether a future iteration of this study is to provide a panel of clinicians and experts with WSI, but also IHC, non-invasive imaging data, clinical meta-data and integrate any clinical outcomes data that may be relevant.

Reviewer #2

(Remarks to the Author)

Thank you to the authors for submitting the work described in the manuscript titled "Pathologists' Diagnosis of Melanoma and Melanocytic Tumors: Discordance, Accuracy and Reproducibility Study", and thank you to the Editor for inviting us to review this manuscript. This manuscript is extremely relevant, given the discussions around melanoma mis/under/overdiagnosis. What sets this manuscript apart from previous work done in this field, is the multi-centre design of the study, relatively large study sample and prospective data collection. Furthermore, it includes a pathologist panel of eight highly experienced international (dermato-)pathologists. Lastly, authors will provide access to the largest panel-validated dataset featuring dermoscopic and histopathological images with metadata, which is extremely relevant for future studies, especially those involving automated assessment of melanoma using Artificial Intelligence.

The manuscript is very well-written according to Nature Communication author guidelines, and the methods are clearly explained in a language that is easy to understand for readers not familiar with the topic. After thorough review of the manuscript, we would like to point out the following comments for your consideration:

1. Introduction paragraph 1: please slightly change wording to better match results in manuscript [2] referenced:
- "around 60,000" – suggested to change to - "57,000"
- "with experts predicting nearly 510,000 new cases and about 96,000 deaths annually by 2040" - suggested to change to - "with experts predicting 510,000 new cases and 96,000 deaths in 2040"
2. Section "Consensus and Arbitration Procedures": please re-write second sentence of first paragraph. While the first sentence of this paragraph includes results of the complete consensus agreement, the second sentence includes results of the complete consensus agreement + majority vote. As readers, we think this does not read well and we think it would be interesting to show the results of the majority vote separately as well. The paragraph could end with a sentence combining the two methods and providing the total number, as provided in the current version of the manuscript.
3. Table 2: please change highest rating frequency of ">5" to ">5 (i.e., no majority vote)", to match with Table 3.
4. Section "Quantification of Interrater Variability": please provide p-value for statement "significantly lower than for all other diagnostic categories".
5. Section "Comparison with Local Pathologists": it would be interesting for the reader to know the initial pathological diagnosis (local pathologist) and ground truth of the 11 cases in which the initial pathological diagnosis diverged from the panel-validated ground truth. If these related to melanoma cases, it would also be good to include a few sentences about this in the discussion, especially because of complete expert consensus.
6. Section "Comparison with Local Dermatologist", sentence starting "within these, ..": please confirm the dermatologist and pathologist diagnosis of the remaining 87/221 cases they disagreed on. If these involved non-melanoma diagnosis by the dermatologist but melanoma diagnosis by the pathologist, this should be added to the text.
7. Section "Comparison with Local Dermatologist", sentence starting "Similarly, the expert ..": please confirm the local pathologist and expert pathologists diagnosis of the remaining 90/265 cases they disagreed on. If these involved non-melanoma diagnosis by the local pathologist but melanoma diagnosis by the expert pathologists, this should be added to the text.
8. The discussion lacks thorough comparison with previous studies. Please add.
9. Discussion, paragraph 3, sentence starting "this discrepancy resulted in ..": from our own experience, dermatological "overdiagnosis" is highly influenced by other factors such as patient concern and this is an important point to add to the discussion.

10. Discussion, paragraph 3, sentence starting "Considering that round 25,000 ...": please also comment on number of patients underdiagnosed and therefore, not receiving (appropriate) treatment.
11. Conclusion, second sentence: "This discordance may lead to numerous healthy individuals being misdiagnosed and unnecessarily treated for melanoma". Some patients may not receive appropriate treatment due to mis/underdiagnosis, and this can lead to even worse patient outcomes. Please comment.
12. Ethics Statement and Reporting Standards: Please check whether HREC approval numbers need to be provided.
13. Section "Patient Cohorts, Image Acquisition, and Slide Acquisition", sentence starting "Data on skin lesions ..": this sentence indicates that all 792 cases included in this study were melanoma-suspicious. However, in section "Comparison with local dermatologist", you say that 134 of 221 cases had clinical suspicion of melanoma by the dermatologist, which means the remaining 87 did not have clinical suspicion (see also point 6). Please explain and clarify.
14. Section "Statistics": you state that "A significance level of $\alpha=0.05$ was set for all analyses", but have not provided a p-value in the results section – see comment 4.
15. Discussion, sentence starting "Although our study design enabled...": please reference previous studies (e.g. <https://pubmed.ncbi.nlm.nih.gov/32338293/>, <https://pubmed.ncbi.nlm.nih.gov/26269032/>, <https://pubmed.ncbi.nlm.nih.gov/16911279/>, and <https://pubmed.ncbi.nlm.nih.gov/19404399/>)
16. Section "Comparison with local pathologists", sentence "...although the panel reached a complete consensus, with all experts independently agreeing on the same diagnosis.": The experts provided independent diagnoses and therefore, we wonder whether the term "consensus" and "consensus agreement" used throughout the manuscript is the correct term. In our opinion, "consensus" is achieved by a group communication process, as described by the Delphi Technique (<https://openpublishing.library.umass.edu/pare/article/id/1418/>). In section "Consensus and Arbitration Procedures": you mention the term "majority agreement", which we believe is a term better suited to describe the process used in this study. Please comment.

Reviewer #3

(Remarks to the Author)

This prospective multi-center study examined the concordance of histopathologic diagnosis of melanoma and melanocytic lesions among a group of 8 experience dermatopathologists, finding good concordance overall, but poorer performance among thin and non-invasive melanomas. This is an extremely important finding as image classification models continue to be developed in dermatology as it highlights the critical point of measurement accuracy in supervised learning data. It is well written and anticipated and addressed many of my queries. I have just a few minor comments that could strengthen the paper:

1. Lines 154-7: why was the least experienced pathologist the one tasked with being the one to reclassify lesion, or am I misunderstanding what was done?
2. Lines 244, 286: the use of "overdiagnosis" is too strong and easy to misinterpret. For the methods presented, I agree that this term is correctly used, however for the broader dermatology readership, a differential diagnosis is usually what is given, or a "rule out melanoma" rather than a bottom line diagnosis of "melanoma."
3. Lines 219-23: for the local pathologist reads of nevi reclassified as melanomas, what was the time interval between the reads, and are there any outcome data on these patients? These would certainly speak to the idea of 'overdiagnosis'
4. Low power might have precluded this analysis, but were there any differences if NIM was separated back into MIS vs LM? Clinically and histologically these considered different entities by many
5. There were 147 'other' lesions. What were these diagnoses?
6. How were dysplastic nevi classified? Were these lumped into 'nevi' or as 'other'?

Reviewer #4

(Remarks to the Author)

Version 1:

Reviewer comments:

Reviewer #1

(Remarks to the Author)

Thank you for addressing the points, comments and questions raised in my initial review.

Reviewer #2

(Remarks to the Author)

Thank you to the authors for addressing the Reviewers' comments and sharing the updated manuscript. Overall, the authors have done an excellent job incorporating the feedback and comments into the text. However, a number of points remain unclear which we would like to point out for your consideration:

1. In general, the line numbers mentioned in your response to the comments, don't match the line numbers in the updated

manuscript, which makes it more challenging find the corresponding text. Furthermore, the updated manuscript still includes comments from the first author SH, stating references to be added to the text. Therefore, I wonder whether the correct manuscript version has been shared with the Reviewers. Please comment.

2. I would like to know how the authors will “provide access to the prospectively and consecutively collected, panel-validated dataset utilized in our study to facilitate further skin cancer research”.

3. In reference to reviewer #2 original comment #6: Local pathologist disagreed with dermatologist in 221/792 cases. For 188/221 cases (134 + 42 + 12), more details are provided. Please also provide more details about remaining 33/221 cases with disagreement.

4. In reference to reviewer #2 original comment #7: expert pathologist panel disagreed with dermatologist in 264/792 cases. For 212/264 cases (168 + 35 + 9), more details are provided. Please also provide more details about remaining 52/264 cases with disagreement.

5. Line 260: why have these numbers changed when no reviewer commented on these numbers? Were the initial numbers (265/792 and 175/265) added by mistake?

6. Line 252, line 256-257, line 262: avoid adding sentences like: “highlighting the extend of avoidable biopsies”, “both indicating potential underdiagnosis by dermatologists”, and “indicating numerous avoidable biopsies as well” in results section. Only present results but don’t interpret results. Interpretation of results is part of Discussion.

7. Line 255-257: “underdiagnosis by dermatologist”. These 12 lesions were excised due to suspicion for malignancy/melanoma. Regardless of melanoma type, thickness, stage etc. it was correctly identified as melanoma. Melanoma stage is challenging to determine based on dermoscopy alone. Therefore, I wonder whether it is fair to say these cases were “underdiagnosed” by the dermatologist. Please comment. Last part of this last sentence “..both indicating potential underdiagnosis by dermatologists” should be removed regardless (see previous comment 6).

8. In reference to reviewer #2 original comment #15: you have stated you added the suggested references to the text. However, I cannot see these references in the text nor in the reference list. Comments from SH in the manuscript indicate these references were intended to be added but have not been added. Please comment.

9. Line 311-312: “...dermatological overdiagnosis (i. e., avoidable biopsies)”. Please carefully review the definition of “overdiagnosis”. Overdiagnosis is the diagnosis of disease that will never cause symptoms or death during a patient's ordinarily expected lifetime (i.e. malignancy correctly identified but unlikely to progress). On the other hand, “misdiagnosis” refers to the incorrect diagnosis of disease (i.e. malignancy identified when in fact non-malignant). Does the “overdiagnosis” of 68 cases of IM/NIM in line 317 refer to “overdiagnosis” or “misdiagnosis”? If these 68 cases were relabelled as naevus or “other”, then the term “misdiagnosis” is better suited here. Please double check and make sure correct terminology is used throughout text, because in line 331 and conclusion the term “misdiagnosis” is used and it is confusing.

10. Line 355-357: the exact same thing was mentioned in lines 331-332, followed by a comment saying this is not feasible. Making a recommendation that is likely unfeasible does not have a lot of power. Please reconsider inclusion of lines 355-357.

Reviewer #3

(Remarks to the Author)

The authors have satisfactorily addressed my comments. I applaud them for showing real data on discrepancies among experienced pathologists, highlighting not just a theoretical risk of hardwiring errors into deep learning tools but the reality of it. As AI continues to expand in medicine, it is critical to understand the quality of training data, and and which data are more likely to be incorrect. A recurring theme in studies such as this using WSI to aid in diagnosis is the focus on the top-n diagnoses vs. ground truth. With the ongoing controversy of melanoma 'overdiagnosis' it is important always to keep in mind, though, that the proper management of a lesion is the more important metric. That is, are we removing lesions that are likely to be harmful, and leaving lesions that will not be? The authors found the greatest discordance among non-invasive and pT1a melanomas. If these lesions are still being removed with appropriate margins, the impact of misdiagnosis is negligible. If, however, these are being mislabeled such that there is no further removal, there would be an expected increase in the risk of recurrence, metastasis, and mortality. As the authors note, patients frequently will push for complete excision of lesions that are equivocal (lines 303-305). Indeed there has been some effort recently to move away from a binary benign/malignant categorization for melanocytic lesions to a more numeric risk scale (<https://pmc.ncbi.nlm.nih.gov/articles/PMC9905960/>). This sort of output would be easy to calculate using deep learning models, though again the question of the accuracy of outcome data arises. It would be unethical to simply monitor a lesion for which there is sufficient concern for melanoma, just as it would be to excise with margins one that is unequivocally benign. Rather, the slow iterative process of tuning diagnostic criteria is more likely to settle on the true risk for any given lesion. This study highlights an area where further investigation is needed for these refinements.

Reviewer #4

(Remarks to the Author)

We thank all reviewers for their insightful questions and their constructive comments. Thank you for the positive feedback on our study addressing “extremely relevant” (R2) and “important work” (R1) that “addresses a foundational issue at the heart of AI capabilities” (R1). We also appreciate that our paper was found to be “well written” (R3) and “easy to understand for readers” (R2), and acknowledge that the “multicentre design of the study, relatively large study sample and prospective data collection” (R2) was valued as strength of our study.

Reviewer #1 - Melanoma, imaging, AI (Remarks to the Author):

Congratulations and thank you to the authors and team for this important work that addresses a foundational issue at the heart of Ai capabilities as applied to pigmented lesion (and in fact more broadly to skin lesion) classification.

The team here expands on the broadly-recognized concept pertaining to the question of the quality of the gold standard of histopathologic diagnosis of melanocytic lesions. It has been well-demonstrated that there is heterogeneity in both inter and intra-reader histopathologic diagnosis, that raises questions about the implications for work in the development of AI tools and capabilities that rely on this gold standard labeling. Further, this topic and issue further plays into current controversies about diagnostic shift and the role of histopathologic diagnosis may play in the rising incidence of melanoma and melanoma-in-situ diagnosis in the absence of rising mortality (i.e. the question of an epidemic driven by over diagnosis/over-surveillance)?

This study further adds to this body of literature with important additions given the robust study design that looks across multiple centers, separates out invasive from in-situ melanoma, and utilized a prospective design, in addition to the valuable sharing of a robust prospectively-collected multi-modal dataset consisting of deeply clinically annotated dermoscopic and histopathologic images.

1. In terms of study design, how does the review of WSI, while increasingly prevalent, differ from usual practice of the local dermatopathologist who may have access to the entire block and additional slides and special stains? In these cases, were only H&E sections reviewed, or was there use of special stains (PRAME, etc?).

Thank you for this thoughtful question. In our study, digitized H&E reference slides for all lesions were displayed in a web-based environment, along with patient-specific (e.g., year of birth, age at diagnosis) and lesion-specific metadata (e.g., lesion localization; see *Methods section*). The online platform was pre-evaluated by a group of reviewers (n=3) and was found to be comparable to, or even preferable to, traditional microscopic slide analysis.

No special stains, additional slides, or physical blocks were available for review (see *Limitations paragraph* in the *Discussion section*). The local pathologists sent the H&E slide(s) that guided their diagnosis to the DKFZ, ensuring that the same decision basis was used by the reviewers in the study. This slide(s) typically represent(s) the region with the greatest tumor thickness, capturing the most aggressive tumor area or cell clone.

2. Were the panelists restricted to pre-specified categories of histopathologic diagnosis? What categories are captured in the "other" vs. nevus category? In other words, how were atypical spitz, MELTUMP, mild/mod/severely atypical nevi, etc. categorized--all together in "other"?

Thank you for your insightful question. The panelists were restricted to pre-specified categories that were defined in collaboration with the most experienced dermatopathologist (Heinz Kutzner).

The "other" category includes a variety of diagnoses, including:

- Actinic keratosis
- Squamous cell carcinoma/Spinalioma
- Basal cell carcinoma
- Benign keratoses (e.g., seborrheic keratosis, lichen-planus-like keratosis)
- Dermatofibroma
- Vascular lesions (e.g., hemangioma)
- Miscellaneous (e.g., Merkel cell carcinoma)

The "nevi" category comprises the following subcategories:

- Spitz nevi and variants
- Dysplastic nevi/Clark nevi
- Acral nevi with palmar-plantar localization
- Recurrent nevi
- Blue nevi
- Combined nevi
- Other types of nevi

Additionally, atypia was recorded separately, in conjunction with the specific nevus subtype. A detailed table listing these subcategories can now be found in the supplementary table 3 (see lines 758 to 761).

3. Very interesting insight that institution 8 has a number of structural factors pertaining to their dermatopathology practice that looks to promote diagnoses more concordant to expert consensus, specifically use of non-invasive imaging capabilities wielded in clinical setting, use of immunohistochemistry, and consensus conference/4 eyes principle. Would be interested in which if these approaches is used at the other institutions (non-invasive diagnostics clinically, special staining, and consensus conference or approach for challenging lesions or IMs).

Thank you for your interest in the diagnostic practices across the participating institutions. Unlike hospital 8, the other clinics typically reserve special staining techniques for borderline cases rather than using them routinely. The four-eyes principle is typically employed when the pathologist faces diagnostic uncertainties, and additional non-invasive diagnostics, such as confocal microscopy, are not widely implemented in their clinical settings.

4. What was the nature of the comparison of pathologist with dermatologist? Top 1? Top 3? Free text or choosing from pre-specified list? How might a situation, for instance, of a clinician who was moderately concerned about MIS but top 1 diagnosis was moderately or severely atypical nevus be treated? Top 1 being atypical nevus but low confidence because Top 2 would include MIS? Wondering whether the approach of concern for melanoma (very high, high, moderate, low, very low) might be a more appropriate approach.

Thank you for your thoughtful questions. The comparison between pathologists and dermatologists was based on the Top 1 diagnosis, chosen from a pre-specified list. For further details please refer to the newly added table 3 in the supplementary materials.

Since only melanoma-suspicious lesions were included in this study (see *Methods section*), all cases inherently involved a high concern for IMs or NIMs. Therefore, the second part of your comment does not apply, as this level of concern was consistent across all cases.

5. It looks like a standardized constructed was created to capture clinical information; was this same information provided in the same way to local pathologist, or was this just for the expert group? Wondering whether these difference may have confounded the differences between local and expert pathologic diagnoses. Appreciate the authors acknowledgment of these differences and variations in the manuscript section on limitations. I believe that there are also significant variations in dermpath practices geographically and between academic institutions and private practice, perhaps noting these additional limitations of the broader applicability of these findings might be appropriate.

Thank you for your question. Both the expert panel and the local pathologists had access to the same clinical information, although it was presented in different formats. The expert group received this information through the online environment used for this study (see supplementary figure 1), while the local pathologists accessed it as part of their routine clinical workflow (e. g., via request forms or patient management systems, depending on the clinic).

We now acknowledge in the limitations section that these differences may have contributed to variations in diagnostic outcomes (see lines 413 and 414). Additionally, we recognize that there are significant variations in dermatopathology practices across geographic regions, as well as between academic institutions and private practice, which may impact the broader applicability of these findings (see lines 414 to 418).

6. The framing of "Comparison with Local Dermatologist" is not an apples to apples (clinical diagnosis with path diagnosis) and may be better converted as the concordance between clinician impression and histopathologic diagnosis. As we know, the diagnosis is one of clinical-pathologic correlation where the two are complementary approaches. I would expect and how that there would be considerable difference in a top 1 clinical dx and pathologic diagnosis where the certainty of the diagnosis moves with concordance with pathologic diagnosis. What I think would be more interesting is level of suspicion or certainty of diagnosis of melanoma/MIS.

Thank you for your thoughtful comments. We agree that comparing clinical impressions with histopathologic diagnoses does not provide a direct one-to-one correlation, as these are complementary approaches to melanoma diagnosis.

In our study, our goal was to assess the concordance between initial clinical suspicion and subsequent histopathologic findings to highlight potential areas of divergence, such as avoidable biopsies. Accordingly, we have changed the heading from “Comparison with Local Dermatologist” to “Concordance with Local Dermatologist’s Clinical Impression” for clarity (see line 269).

We also appreciate your point regarding the importance of diagnostic certainty. To address this, we have already included table 5, which outlines the association between dermatologists’ level of diagnostic confidence in their own clinical impression and the corresponding histopathologic diagnosis (see lines 302 to 314).

7. Page 17 last paragraph speaks to the discordance between local read and expert read with the majority of instances representing an "overcall" from local pathologist. Perhaps would soften the language re: "substantial overdiagnosis" and perhaps just have the proportion of over/under call stand alone and the line about "suggestion that approximately 10% of all melanoma cases might be pathologically overdiagnosed". I would also argue that deserving similar highlighting is the proportion of cases that may be pathologically under diagnosed though less frequent or common.

Thank you for this important observation. We agree and have softened the language by removing the statement suggesting that approximately 10% of all melanoma cases might be pathologically overdiagnosed, as well as the nationwide projection based on this estimate (see line 359). Additionally, we have highlighted the implications of cases that may be pathologically underdiagnosed, to ensure a balanced discussion of both over- and underdiagnosis (see lines 371 to 373).

8. I am curious as to whether the authors may perform an analysis that might support their recommendation of 2 pathologist independently assess a case (i.e. what would the data have looked like randomly choosing a pair of the expert reviewer panel vs. the full 8)? How does the practice of local consensus conference or 4 eye approach potentially compare or address this in the setting of resource constraint?

Thank you for this thoughtful suggestion. Retrospectively, randomly selecting the diagnostic votes of two reviewers from our panel for an additional analysis would not align with our recommendation, which involves independent review by two pathologists, followed by the opportunity for additional staining or consultation of a third party in cases of discordance (see lines 373 to 376). However, we are considering a prospective follow-up study with two arms — one following routine procedures and the other incorporating a “four-eye” review process, with a digital panelist option. This approach would allow us to further explore how such a protocol could enhance diagnostic accuracy in real-life, resource-limited clinical settings. While this analysis is outside the scope of the current paper, we appreciate your suggestion and will consider it for future research.

9. Appreciate the paradox called out in the discussion section where we have a real-world clinical opportunity of generating more consistent and accurate pathologic diagnoses and AI can potentially serve as a technological layer to help, but that is predicated on more robust ground-truth labeling. I wonder whether a future iteration of this study is to provide a panel of clinicians and experts with WSI, but also IHC, non-invasive imaging data, clinical meta-data and integrate any clinical outcomes data that may be relevant.

Thank you for highlighting this important point. We are indeed currently working on a follow-up study designed that will provide an expert panel with WSIs, IHC stainings (e.g., MelanA), original dermoscopic images, and clinical metadata. By integrating these diverse data layers, we aim to further enhance diagnostic consistency and accuracy, ultimately contributing to a reevaluation of how ground truth should be established in both routine patient care and future AI research.

Reviewer #2 - melanoma pathology (Remarks to the Author):

Thank you to the authors for submitting the work described in the manuscript titled “Pathologists’ Diagnosis of Melanoma and Melanocytic Tumors: Discordance, Accuracy and Reproducibility Study”, and thank you to the Editor for inviting us to review this manuscript. This manuscript is extremely relevant, given the discussions around melanoma mis/under/overdiagnosis. What sets this manuscript apart from previous work done in this field, is the multi-centre design of the study, relatively large study sample and prospective data collection. Furthermore, it includes a pathologist panel of eight highly experienced international (dermato-)pathologists. Lastly, authors will provide access to the largest panel-validated dataset featuring dermoscopic and histopathological images with metadata, which is extremely relevant for future studies, especially those involving automated assessment of melanoma using Artificial Intelligence.

The manuscript is very well-written according to Nature Communication author guidelines, and the methods are clearly explained in a language that is easy to understand for readers not familiar with the topic. After thorough review of the manuscript, we would like to point out the following comments for your consideration:

1. Introduction paragraph 1: please slightly change wording to better match results in manuscript [2] referenced:

- “around 60,000” – suggested to change to - “57,000”
- “with experts predicting nearly 510,000 new cases and about 96,000 deaths annually by 2040” - suggested to change to - “with experts predicting 510,000 new cases and 96,000 deaths in 2040”

Thank you for your suggestion. We have updated the wording in the *Introduction paragraph* to better align with the referenced manuscript [2] (see lines 100 and 103).

2. Section “Consensus and Arbitration Procedures”: please re-write second sentence of first paragraph. While the first sentence of this paragraph includes results of the complete consensus agreement, the second sentence includes results of the complete consensus agreement + majority vote. As readers, we think this does not read well and we think it would be interesting to show the results of the majority vote separately as well. The paragraph could end with a sentence combining the two methods and providing the total number, as provided in the current version of the manuscript.

Thank you for your feedback. We have updated the *Consensus and Arbitration Procedures section* to separately present the complete agreement and majority vote results, followed by a combined total (see lines 182 to 186).

3. Table 2: please change highest rating frequency of “>5” to “>5 (i.e., no majority vote)”, to match with Table 3.

Thank you for your careful observation. We have updated table 2 accordingly to ensure consistency with table 3.

4. Section “Quantification of Interrater Variability”: please provide p-value for statement “significantly lower than for all other diagnostic categories”.

Thank you for your comment. We determined the significant difference using the percentile method, noting that the corresponding 95% confidence intervals do not overlap at any point. This clarification has been added to lines 224 and 511 to 513 of the *Results and Statistics section* to further explain our procedure.

5. Section “Comparison with Local Pathologists”: it would be interesting for the reader to know the initial pathological diagnosis (local pathologist) and ground truth of the 11 cases in which the initial pathological diagnosis diverged from the panel-validated ground truth. If these related to melanoma cases, it would also be good to include a few sentences about this in the discussion, especially because of complete expert consensus.

Thank you for your suggestion. We have added the initial pathological diagnoses alongside the panel-validated ground truths for the cases where diagnoses diverged in the *Comparison with Local Pathologists section* (see lines 250 to 254). Additionally, we have expanded the *Discussion section* to address these cases, as requested (see lines 356 to 359).

6. Section “Comparison with Local Dermatologist”, sentence starting “within these, ..”: please confirm the dermatologist and pathologist diagnosis of the remaining 87/221 cases they disagreed on. If these involved non-melanoma diagnosis by the dermatologist but melanoma diagnosis by the pathologist, this should be added to the text.

Thank you for your thoughtful comment. We have confirmed the diagnoses for the remaining cases and expanded the *Comparison with Local Dermatologist section* to consider instances where dermatologists diagnosed non-melanoma, while pathologists diagnosed melanoma (see lines 274 to 278).

7. Section “Comparison with Local Dermatologist”, sentence starting “Similarly, the expert ..”: please confirm the local pathologist and expert pathologists diagnosis of the remaining 90/265 cases they disagreed on. If these involved non-melanoma diagnosis by the local pathologist but melanoma diagnosis by the expert pathologists, this should be added to the text.

Thank you for your insightful comment. We have confirmed the diagnoses for the remaining cases and expanded the *Comparison with Local Dermatologist section* to include instances where dermatologists diagnosed non-melanoma, while the expert pathologists diagnosed melanoma (see lines 295 to 300).

8. The discussion lacks thorough comparison with previous studies. Please add.

Thank you for your valuable feedback. We have expanded the *Discussion section* to include a thorough comparison with previous studies, contextualizing our findings within the existing body of research (see lines 332 to 341).

9. Discussion, paragraph 3, sentence starting “this discrepancy resulted in ..”: from our own experience, dermatological “overdiagnosis” is highly influenced by other factors such as patient concern and this is an important point to add to the discussion.

Thank you for this important observation. We agree that factors such as patient concern can significantly contribute to dermatological “overdiagnosis.” We have expanded the *Discussion section* accordingly (see lines 349 to 352), emphasizing that overdiagnosis rates should be interpreted with caution, giving these additional influences on diagnostic decisions.

10. Discussion, paragraph 3, sentence starting “Considering that round 25,000 ...”: please also comment on number of patients underdiagnosed and therefore, not receiving (appropriate) treatment.

Thank you for highlighting this point. We have addressed the concern of underdiagnosis in the *Discussion section*, noting that numerous patients may not receive timely and appropriate treatment (see lines 371 to 373). However, determining exact numbers remains challenging due to a high “*Dunkelziffer*” (i.e., the unknown number of unreported or undetected cases).

11. Conclusion, second sentence: “This discordance may lead to numerous healthy individuals being misdiagnosed and unnecessarily treated for melanoma”. Some patients may not receive appropriate treatment due to mis/underdiagnosis, and this can lead to even worse patient outcomes. Please comment.

Thank you for your thoughtful comment. We have expanded the *Conclusion section* underscoring that misdiagnosis and underdiagnosis can also lead to some patients not receiving appropriate treatment, potentially resulting in worse outcomes (see lines 425 to 427).

12. Ethics Statement and Reporting Standards: Please check whether HREC approval numbers need to be provided.

Thank you for your comment. We have included the HREC approval numbers in the *Ethics Statement and Reporting Standards section* (see lines 434 to 437).

13. Section “Patient Cohorts, Image Acquisition, and Slide Acquisition”, sentence starting “Data on skin lesions ..”: this sentence indicates that all 792 cases included in this study were melanoma-suspicious. However, in section “Comparison with local dermatologist”, you say that 134 of 221 cases had clinical suspicion of melanoma by the dermatologist, which means the remaining 87 did not have clinical suspicion (see also point 6). Please explain and clarify.

Thank you for pointing this out. To clarify, all 792 cases included in the study were deemed clinically suspicious for melanoma, either as a primary diagnosis or as a secondary/differential diagnosis intended to rule out melanoma. We have clarified this distinction in the *Methods section* (see lines 446 and 447).

In the *Comparison with Local Dermatologist section*, the label always reflects the clinical primary diagnosis.

14. Section “Statistics”: you state that “A significance level of $\alpha=0.05$ was set for all analyses”, but have not provided a p-value in the results section – see comment 4.

Thank you for your careful observation. In this context, the significance level of $\alpha=0.05$ pertains to the calculation of 95% (1 - α) confidence intervals. Significant differences between k values were determined using the percentile method, as noted in comment 4. We have clarified this in lines 511 to 513 of the *Statistics section*.

15. Discussion, sentence starting “Although our study design enabled...”: please reference previous studies (e.g. <https://pubmed.ncbi.nlm.nih.gov/32338293/>, <https://pubmed.ncbi.nlm.nih.gov/26269032/>, <https://pubmed.ncbi.nlm.nih.gov/16911279/>, and <https://pubmed.ncbi.nlm.nih.gov/19404399/>)

Thank you for your feedback. We have added the suggested references to the *Discussion section* (see lines 406 and 408).

16. Section “Comparison with local pathologists”, sentence “..although the panel reached a complete consensus, with all experts independently agreeing on the same diagnosis.”: The experts provided independent diagnoses and therefore, we wonder whether the term “consensus” and “consensus agreement” used throughout the manuscript is the correct term. In our opinion, “consensus” is achieved by a group communication process, as described by the Delphi Technique (<https://openpublishing.library.umass.edu/pare/article/id/1418/>). In section “Consensus and Arbitration Procedures”: you mention the term “majority agreement”, which we believe is a term better suited to describe the process used in this study. Please comment.

Thank you for this valuable observation. We agree that “majority/complete agreement” more accurately reflects the process used in our study, as it was based on independent diagnoses rather than a group communication process. Accordingly, we have replaced the term “consensus” with “majority/complete agreement” throughout the entire manuscript.

Reviewer #3 - melanoma diagnostics (Remarks to the Author):

This prospective multi-center study examined the concordance of histopathologic diagnosis of melanoma and melanocytic lesions among a group of 8 experience dermatopathologists, finding good concordance overall, but poorer performance among thin and non-invasive melanomas. This is an extremely important finding as image classification models continue to be developed in dermatology as it highlights the critical point of measurement accuracy in supervised learning data. It is well written and anticipated and addressed many of my queries. I have just a few minor comments that could strengthen the paper:

1. Lines 154-7: why was the least experienced pathologist the one tasked with being the one to reclassify lesion, or am I misunderstanding what was done?

Thank you for your insightful question. To clarify, CNG, an independent reviewer not part of the expert panel, was tasked exclusively with determining the BT category of lesions flagged by the expert panel for reclassification (n=13). Selecting CNG added an extra layer of validation, ensuring that histopathological features supporting the reclassification were consistently observed. We have now clarified this in lines 160 to 163 and 234 to 237.

2. Lines 244, 286: the use of "overdiagnosis" is too strong and easy to misinterpret. For the methods presented, I agree that this term is correctly used, however for the broader dermatology readership, a differential diagnosis is usually what is given, or a "rule out melanoma" rather than a bottom line diagnosis of "melanoma."

Thank you for this thoughtful comment. We recognize that "overdiagnosis" may imply a definitive diagnosis, which could be misinterpreted by a broader dermatology audience. Our intent was to highlight avoidable biopsies rather than a confirmed melanoma diagnosis. We have adjusted the wording in lines 273 and 295 accordingly.

3. Lines 219-23: for the local pathologist reads of nevi reclassified as melanomas, what was the time interval between the reads, and are there any outcome data on these patients? These would certainly speak to the idea of 'overdiagnosis'

Thank you for your question. The reclassification of nevi as melanomas was conducted solely within the context of this study. Due to data protection restrictions, we were not allowed to recontact patients or gather follow-up data that might have further supported the expert panel's reclassification.

4. Low power might have precluded this analysis, but were there any differences if NIM was separated back into MIS vs LM? Clinically and histologically these considered different entities by many

Thank you for your thoughtful question. We conducted separate analyses for NIMs, distinguishing MIS from LM, and found no significant differences. To maintain higher statistical power, we have therefore kept the grouped results in the manuscript.

5. There were 147 'other' lesions. What were these diagnoses?

Thank you for your thoughtful question. The category of "other" lesions (n=147) comprises a variety of diagnoses, including:

- Actinic keratosis
- Squamous cell carcinoma/Spinalioma
- Basal cell carcinoma
- Benign keratoses (e.g., seborrheic keratosis, lichen-planus-like keratosis)
- Dermatofibroma
- Vascular lesions (e.g., hemangioma)
- Miscellaneous (e.g., merkel cell carcinoma)

A detailed table listing these subcategories can now be found in the supplementary table 3 (see lines 758 to 761).

6. How were dysplastic nevi classified? Were these lumped into 'nevi' or as 'other?'

Thank you for raising this question. Dysplastic nevi were included within the "nevi" category. Further details can be found in the supplementary materials, where we have added table 3 for clarification (lines 758 to 761).

.

.

Reviewer #4 - Early Career Researcher (Remarks to the Author):

Thank you for your efforts in co-reviewing our manuscript. Your insights have been invaluable, and we appreciate the time and thought you contributed to the review process.

Reviewer #2 (Remarks to the Author):

Thank you to the authors for addressing the Reviewers' comments and sharing the updated manuscript. Overall, the authors have done an excellent job incorporating the feedback and comments into the text. However, a number of points remain unclear which we would like to point out for your consideration:

1. In general, the line numbers mentioned in your response to the comments, don't match the line numbers in the updated manuscript, which makes it more challenging find the corresponding text. Furthermore, the updated manuscript still includes comments from the first author SH, stating references to be added to the text. Therefore, I wonder whether the correct manuscript version has been shared with the Reviewers. Please comment.

Thank you for bringing this to our attention. We sincerely apologize for the discrepancy between the line numbers in our response letter and the updated manuscript. This was not intentional, and we have ensured that all references are now correctly cited in the text and the reference list.

2. I would like to know how the authors will "provide access to the prospectively and consecutively collected, panel-validated dataset utilized in our study to facilitate further skin cancer research".

Thank you for your question regarding data access. The prospectively and consecutively collected, panel-validated dataset utilized in this study will be made available for non-commercial skin cancer research. External researchers may apply for access through the SCP Data Protection Committee, which reviews applications based on alignment with patient consent, valid ethics approval, and other relevant criteria. Commercial use of the dataset is strictly prohibited (see data availability statement).

3. In reference to reviewer #2 original comment #6: Local pathologist disagreed with dermatologist in 221/792 cases. For 188/221 cases (134 + 42 + 12), more details are provided. Please also provide more details about remaining 33/221 cases with disagreement.

Thank you for your thoughtful comment. We have confirmed the diagnoses for the remaining 33 cases and expanded the *Comparison with Local Dermatologist* section to include these details.

4. In reference to reviewer #2 original comment #7: expert pathologist panel disagreed with dermatologist in 264/792 cases. For 212/264 cases (168 + 35 + 9), more details are provided. Please also provide more details about remaining 52/264 cases with disagreement.

We appreciate your observation. We have confirmed the diagnoses for the remaining 52 cases and updated the *Comparison with Local Dermatologist* section accordingly.

5. Line 260: why have these numbers changed when no reviewer commented on these numbers? Were the initial numbers (265/792 and 175/265) added by mistake?

Thank you for pointing out this discrepancy. Upon reviewing the SPSS analysis, we identified a typographical error in our initial manuscript. The data has been recalculated and corrected. Please note that the primary findings and trends of the study remain unchanged.

6. Line 252, line 256-257, line 262: avoid adding sentences like: “highlighting the extend of avoidable biopsies”, “both indicating potential underdiagnosis by dermatologists”, and “indicating numerous avoidable biopsies as well” in results section. Only present results but don’t interpret results. Interpretation of results is part of Discussion.

We appreciate your feedback. The sentences you highlighted have been removed to ensure the Results section focuses solely on presenting findings. Interpretations have been shifted to the Discussion section.

7. Line 255-257: “underdiagnosis by dermatologist”. These 12 lesions were excised due to suspicion for malignancy/melanoma. Regardless of melanoma type, thickness, stage etc. it was correctly identified as melanoma. Melanoma stage is challenging to determine based on dermoscopy alone. Therefore, I wonder whether it is fair to say these cases were “underdiagnosed” by the dermatologist. Please comment. Last part of this last sentence “..both indicating potential underdiagnosis by dermatologists” should be removed regardless (see previous comment 6).

Thank you for pointing this out. We agree that the term "underdiagnosis" may not be appropriate in this context. The last part of the sentence has been removed, as recommended, to avoid ambiguity and misinterpretation.

8. In reference to reviewer #2 original comment #15: you have stated you added the suggested references to the text. However, I cannot see these references in the text nor in the reference list. Comments from SH in the manuscript indicate these references were intended to be added but have not been added. Please comment.

We sincerely apologize for this oversight. The suggested references have now been added to the relevant sections in the text and included in the reference list.

9. Line 311-312: “...dermatological overdiagnosis (i. e., avoidable biopsies)”. Please carefully review the definition of “overdiagnosis”. Overdiagnosis is the diagnosis of disease that will never cause symptoms or death during a patient’s ordinarily expected lifetime (i.e. malignancy correctly identified but unlikely to progress). On the other hand, “misdiagnosis” refers to the incorrect diagnosis of disease (i.e. malignancy identified when in fact non-malignant). Does the “overdiagnosis” of 68 cases of IM/NIM in line 317 refer to “overdiagnosis” or “misdiagnosis”? If these 68 cases were relabelled as naevus or “other”, then the term “misdiagnosis” is better suited here. Please double check and make sure correct terminology is used throughout text, because in line 331 and conclusion the term “misdiagnosis” is used and it is confusing.

Thank you for raising this important distinction. We have carefully reviewed and revised the manuscript to ensure the terms "overdiagnosis" and "misdiagnosis" are used accurately and consistently throughout the text.

10. Line 355-357: the exact same thing was mentioned in lines 331-332, followed by a comment saying this is not feasible. Making a recommendation that is likely unfeasible does not have a lot of power. Please reconsider inclusion of lines 355-357.

We appreciate your feedback on this recommendation. Upon reflection, we have removed lines 355–357 from the manuscript, as including recommendations that are not practically feasible would not strengthen our conclusions.